# Trends of cleft surgeries and predictors of late primary surgery among children with cleft lip and palate at the University College Hospital, Nigeria: A retrospective cohort study

Afieharo Igbibia Michael [1,2]* , Gbenga Olorunfemi[3], Adeola Olusanya[4‡], Odunayo Oluwatosin[1,2‡]

1 Department of Surgery, College of Medicine, University of Ibadan, Ibadan, Nigeria, 2 Department of Plastic, Reconstructive and Aesthetic Surgery, University College Hospital, Ibadan, Nigeria, 3 Division of Epidemiology and Biostatistics, School of Public Health, University of Witwatersrand, Johannesburg, South Africa, 4 Department of Oral and Maxillofacial Surgery, Faculty of Dentistry, University of Ibadan, Ibadan, Nigeria

☯ These authors contributed equally to this work.
‡ AO and OO also contributed equally to the work.
* afiemichael@gmail.com

**Data Availability Statement:** Due to third-party restrictions on data availability from the Smile Train database data cannot be uploaded as a Supporting

## Abstract

### Background

Cleft of the lip and palate is the most common craniofacial birth defect with a worldwide incidence of one in 700 live births. Early surgical repairs are aimed at improving appearance, speech, hearing, psychosocial development and avoiding impediments to social integration. Many interventions including the Smile Train partner model have been introduced to identify and perform prompt surgical procedures for the affected babies. However, little is known about the trends of the incidence and surgical procedures performed at our hospital. Nothing is also known about the relationship between the clinical characteristics of the patients and the timing of primary repairs.

### Objective

To determine the trends in cleft surgeries, patterns of cleft surgeries and identify factors related to late primary repair at the University College Hospital, UCH, Ibadan, Nigeria.

### Methods

A retrospective cohort study and trends analysis of babies managed for cleft lip and palate from January 2007 to January 2019 at the UCH, Ibadan was conducted. The demographic and clinical characteristics were extracted from the Smile Train enabled cleft database of the hospital. The annual trends in rate of cleft surgeries (number of cleft surgeries per 100,000 live births) was represented graphically. Chi square test, Student's t-test and Mann Whitney U were utilised to assess the association between categorical and continuous variables and delay in cleft surgery ($\geq$12 months for lip repair, $\geq$18months for palatal repair).

Information file. Data may be provided on request to the Smile Train data manager (stxadmin@smiletrain.org). The authors reasonably believe that Smile Train will grant access for fair and ethical utility of their data for research related purposes. The authors did not have special access or request privileges.

**Funding:** The author(s) received no specific funding for this work.

**Competing interests:** The authors have declared that no competing interests exist.

Kaplan-Meier graphs with log-rank test was used to examine the association between socio-demographic variables and the outcome (late surgery). Univariable and multivariable Cox proportional hazard regression was conducted to obtain the hazard or predictors of delayed cleft lip surgery. Stata version 17 (Statacorp, USA) statistical software was utilised for analysis.

## Results

There were 314 cleft surgeries performed over the thirteen-year period of study. The male to female ratio was 1.2:1. The mean age of the patients was 58.08 ± 99.65 months. The median age and weight of the patients were 11 (IQR:5–65) months and 8 (IQR: 5.5–16) kg respectively. Over half (n = 184, 58.6%) of the cleft surgeries were for primary repairs of the lip and a third (n = 94, 29.9%) were surgeries for primary repairs of the palate. Millard's rotation advancement flap was the commonest lip repair technique with Fishers repair introduced within two years into the end of the study. Bardachs two flap palatoplasty has replaced Von Langenbeck palatoplasty as the commonest method of palatal repair. The prevalence of late primary cleft lip repair was about a third of the patients having primary cleft lip surgery while the prevalence of late palatal repair was more than two thirds of those who received primary palatoplasty. Compared with children who had bilateral cleft lip, children with unilateral cleft lip had a significantly increased risk of late primary repair (Adj HR: 22.4, 955 CI: 2.59–193.70, P-value = 0.005).

## Conclusion

There has been a change from Von Langenbeck palatoplasty to Bardachs two-flap palatoplasty. Intra-velar veloplasty and Fisher's method of lip repair were introduced in later years. There was a higher risk of late primary repair in children with unilateral cleft lip.

## Introduction

Cleft of the lip and palate is the most common craniofacial birth defect with a worldwide incidence of one in 700 live births [1]. This burden is higher in low- and middle-income countries with a higher number of untreated clefts [2]. Early surgical repairs are aimed at improving appearance, speech, hearing, psychosocial development and avoiding impediments to social integration [3]. Some guidelines have been put forth to optimize safety, ensure adequacy of repair, and improve function while limiting morbidity to the patient with the cleft deformity [4, 5]. Surgical procedures should be well planned to reduce exposure to anaesthesia from multiple surgeries [3, 6, 7]. Deciding on a method of cleft lip or palate repair is based on the surgeon's experience, individual patient cleft type peculiarity and evidence based superior outcomes associated with a surgical method. More common methods in cleft lip and palate repair include Millard rotation advancement, Tennison Randall, Fisher's lip repairs, Bardach, Von Langenbeck, Furlow and intravelar veloplasty.

One of the common guidelines for the timing of surgical cleft lip repair is the rule of ten [4]. Although its validity is questioned, the "rule of ten" guideline is still widely used [8–10]. By this rule, children who are ten weeks old, attained the weight of five kilograms and have a haemoglobin of 10g/dl can commence primary lip repair if there are no other mitigating medical

conditions [11]. This rule is also employed in our institution. The American Cleft Palate-Craniofacial Association (ACPA) introduced parameters to evaluate and treat patients with cleft lip and palate [3]. The earlier edition of ACPA guidelines recommended that lip repair be completed by six months and palatal repair by 18 months while the latest revision provides that initiation of lip repair be commenced before 12 months and palatal repair completed by 18 months [3]. The association further advised that earlier surgeries are desired if the conditions are right.

Based on the ACPA guidelines, Cassell et al. from a resourced nation, reported nearly 90% of the patients in their study had had their surgeries by the age of six months [12]. A study by Cornway et al. looked at cleft surgeries across Africa. They reported a mean age of nine years for commencement of primary surgeries [13]. From China an average age of 1.8 years for lip repair and 5.9 years for palatal repair was reported by Kling et al [14]. A study from Northern Nigeria reported a mean age of 12.4 years for the commencement of primary surgeries in their cohort of 149 patients with cleft anomalies [8]. Onah et al also reported a delay in commencement of primary surgeries for cleft of the palate in more than half of their study population in southeast Nigeria [15].

The Smile Train, the world's largest cleft centred charity-based organisation that provides funding for children with clefts began partnership with Nigeria in 2002 aimed at prompt and safe cleft surgery [16]. Nevertheless, some reports of delay in cleft repair still occurred in the country. [8, 15, 17, 18]. The demographic and clinical parameters that may be responsible for delay in primary surgeries have not been fully explored in Nigeria. This lack of information makes it difficult to design innovative methods to mitigate this problem, help the patients attain early repairs and improve their quality of life. A critical review cleft surgical trends and delay patterns in our hospital with an understanding of the demographic and clinical predictors of this delay would inform strategies at curbing these occurrences. Hence, we aimed to know the trends in cleft surgeries, determine the patterns of cleft surgeries and identify factors related to late primary repair.

## Methods

### Study design and setting

This was a retrospective cohort study of all cleft surgeries from January 2007 to December 2019. The study was carried out at the University College Hospital, Ibadan, a tertiary care, federal government owned, facility in South-Western Nigeria. The hospital commenced partnership with the Smile Train in 2007 and has been providing free cleft care to patients with clefts. The rule of 10 as proposed by Wilhelmsen and Musgrave [4] and modified by Millard [8] is used for guiding scheduling of our patients for primary lip repair. The data of patients who received cleft care are entered into the Smile Train Express (STX). This database is specific to the institution. All adult patients and parents of children who received care gave a written informed consent for surgery and for entry into and utilization of their details in this database for research. Anonymised data was extracted from the STX database and crosschecked with existing patient's case record forms and case notes. Data on annual live births were obtained from the hospital records department. The study was approved by the institutional review board. (Ethics review number: NHREC/05/01/2008a). Additional consent for this study was waived by the ethics committee due to the retrospective nature of the study.

### Definition of terms

*Typical cleft deformities* were classified according to the ACPA classification of clefts into clefts of the primary palate and cleft of the secondary palate. Further sub classifications were done in

line with the International Classification of Diseases [5] (ICD) 10 classification into Cleft Lip (CL), which is cleft of the primary palate alone, Cleft lip and palate (CLP), these are clefts that involve both the primary and secondary palates and Cleft palate (CP), which is an isolated cleft of the secondary palate. Further subdivisions used in this study were based on severity of the defect (incomplete/complete), laterality (right/left/bilateral).

*Atypical clefts* were clefts that involved any other part of the face that differed from the commoner pattern of cleft lip and palate. Here they were classified according to Paul Tessier [19].

*Primary surgeries* were defined as the first surgeries performed on any anatomical region of the cleft.

*Secondary surgeries*—Subsequent surgeries performed after the primary surgery on the same anatomical region of the cleft.

## Variables

Primary outcome measure was Late primary surgery (age at surgery $\geq$ 12 months for lip repair and $\geq$ 18 months for palatal repair). Independent variables were age, gender, weight, type of deformity, severity of the deformity, age at surgery and year of surgery. Only primary surgeries of typical clefts were utilised for bivariate and multivariate analysis to determine the primary outcomes. Atypical clefts and secondary surgeries were not analyzed further.

## Data analysis

Categorical variables are presented as frequency and percentages while continuous variables are presented as mean and standard deviation where normally distributed or median and interquartile range if not normally distributed. Yearly trends in rate of cleft surgeries (number of cleft surgeries per 100,000 live births) are presented in graphical form. Chi square test was used to assess the association between sociodemographic characteristics and delay in cleft surgery. Student's t-test and Mann Whitney U were utilised to assess the association between continuous variables and delay in cleft surgery ($\geq$12 months for lip repair, $\geq$18months for palatal repair). Pearson's correlation coefficient was utilised to determine the association between weight and age at surgery. The time-variate variable was "time to cleft lip surgery". Kaplan-Meier graphs with log-rank test was used to examine the association between sociodemographic variables and the outcome (late surgery). Univariable and multivariable Cox proportional hazard regression was conducted to obtain the hazard or predictors of delayed cleft lip surgery. P-value of $\leq$ 0.05 was considered as statistically significant. Stata version 17 (Statacorp, USA) statistical software was utilised for analysis.

## Results

### Demographic and clinical characteristics

There were 314 cleft surgeries performed over the thirteen-year period of study (January 2007-December 2019). Table 1 shows the demographic and clinical characteristics of the patients. The male to female ratio was 1.2:1. The mean age of the patients was 58.08 ± 99.65 months. The median age and weight of the patients were 11 (IQR:5–65) months and 8 (IQR: 5.5–16) kg respectively. About half (n = 167, 51.3%) of the patients were infants (aged $\leq$ 12months) while one-tenth (10.2%, n = 32) of the operated patients were adults. Nearly half of the patients had a primary diagnosis of cleft lip (n = 130/314, 41.4%) and about one-third had cleft of the lip and palate (n = 119/314, 37.9%). Unilateral defects were almost twice as common on the left 99 (63.9%) as on the right 56 (36.1%). Of the 314 performed cleft surgeries, over half (n = 184, 58.6%) were for primary repairs of the lip and a third (n = 94,

**Table 1. Demographic and clinical characteristics of the patients.**

| Variable | Frequency N (%) |
|---|---|
| **Gender (N = 314)** | |
| Female | 173(44.9) |
| Male | 141 (55.1) |
| **Age (Median, IQR), months** | 11(5–65) |
| **Age category (N = 314)** | |
| Infants | 161(51.3) |
| Toddler | 43(13.7) |
| Pre school | 14(4.5) |
| School age | 42(13.4) |
| Adolescent | 22(7.0) |
| Adult | 32(8.9) |
| **Weight (Median, IQR), kg** | 8.0(5.5–16) |
| < 5 | 34(12.2) |
| ≥5 | 244(87.8) |
| **Primary Diagnosis (N = 314)** | |
| Cleft lip | 130 (41.4) |
| Cleft lip and palate | 119(37.9) |
| Cleft palate | 53(16.9) |
| Atypical cleft | 12(3.8) |
| **Type of Surgeries done (N = 314)** | |
| Primary unilateral cleft lip | 155 (49.4) |
| Primary bilateral cleft lip | 29 (9.2) |
| Primary cleft palate | 94(29.9) |
| Primary Atypical cleft | 12 (3.8) |
| Secondary cleft surgery | 24 (7.6) |
| **Severity of the unilateral cleft lip (N = 155)** | |
| Incomplete cleft lip | 55 (35.6) |
| Complete cleft lip | 57 (36.7) |
| Complete cleft lip and palate | 43 (27.7) |
| **Severity of the cleft palate (N = 94)** | |
| Incomplete | 24(25.5) |
| Complete | 70 (74.5) |

29.9%) were surgeries for primary repairs of the palate. Isolated defects of the palate were 50 (53.2%). Within this group was a female predominance of 30 (66.7%). The distribution of primary surgeries between incomplete cleft lip, complete cleft lip and complete cleft lip and palate were fairly even. Primary surgeries for complete cleft palate were more than twice the number for incomplete cleft palate.

## Trends in cleft surgeries

There was a steady rise in the number of cleft surgeries from inception in 2007 with 19 cleft surgeries to a peak of 39 cleft surgeries in 2010 and 2011, Fig 1. A steep drop to no cleft surgeries was seen in 2013. A subsequent steady increase was then seen thereafter. However, the number of surgeries has not reached the initial peak. The cleft lip and palate rate per 1000 live births demonstrates three peaks, in the years 2010, 2014 and 2018 with rates of 20.2/1000, 7.1/1000 and 17.4/1000 respectively, Fig 2, Table 2.

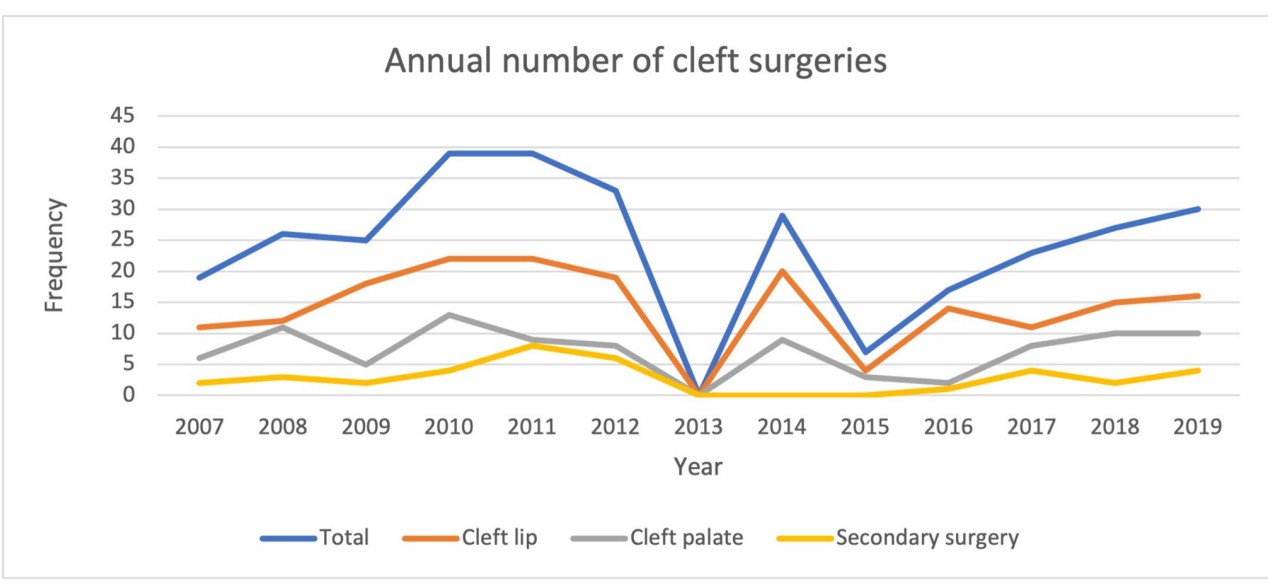

**Fig 1. Annual trends in cleft surgeries.**

## Trends in methods of cleft lip and palate repair

The most common method of cleft lip repair was the variant of Millard's rotation advancement flap, Fig 3. While the classical Millard's repair was used at the outset in 2007 it's use was discontinued in 2016. The least favoured method of cleft lip repair was the rose Thompson

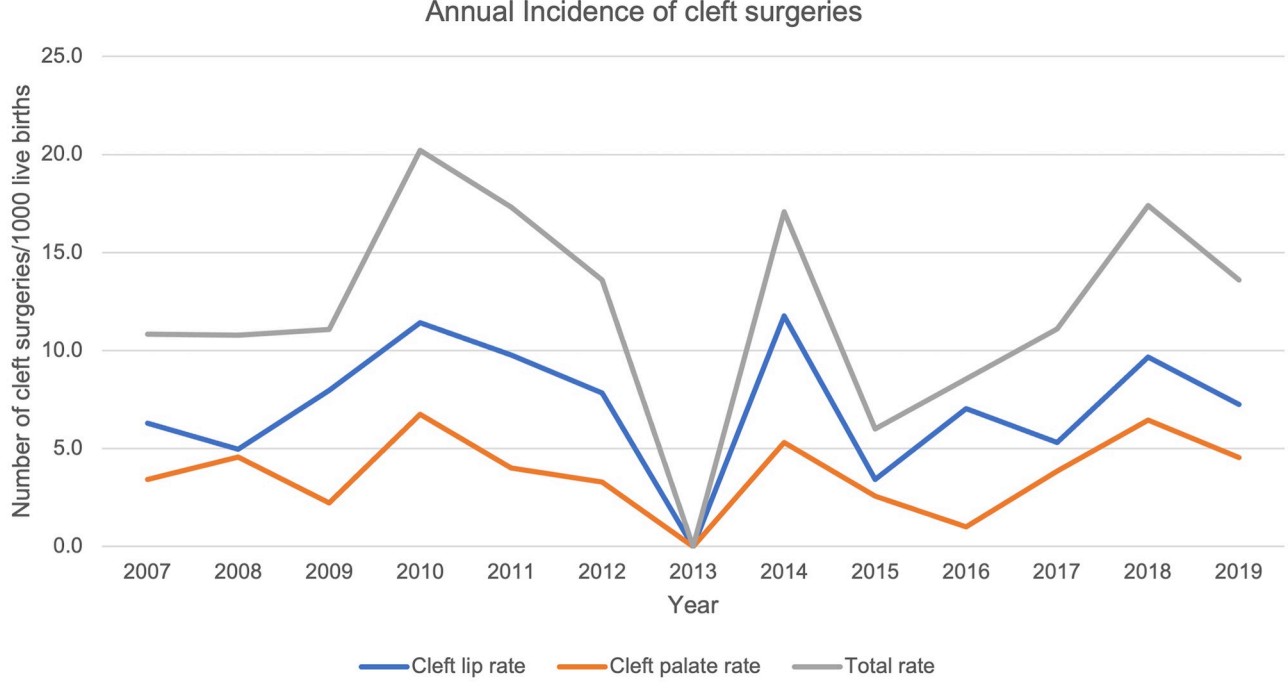

**Fig 2. Annual incidence of cleft surgeries per 1000 live births.**

**Table 2. Trends in the annual number of clefts and incidence (2007–2019).**

| Year | Total | Livebirth | Cleft lip | Cleft lip rate/1000 live births | Cleft palate | Cleft palate rate/1000 live births |
|------|-------|-----------|-----------|--------------------------------|--------------|-----------------------------------|
| 2007 | 19 | 1752 | 11 | 6.3 | 6 | 3.4 |
| 2008 | 26 | 2414 | 12 | 5.0 | 11 | 4.6 |
| 2009 | 25 | 2259 | 18 | 8.0 | 5 | 2.2 |
| 2010 | 39 | 1928 | 22 | 11.4 | 13 | 6.7 |
| 2011 | 39 | 2251 | 22 | 9.8 | 9 | 4.0 |
| 2012 | 33 | 2429 | 19 | 7.8 | 8 | 3.3 |
| 2013 | 0 | 2,077 | 0 | 0.0 | 0 | 0.0 |
| 2014 | 29 | 1,699 | 20 | 11.8 | 9 | 5.3 |
| 2015 | 7 | 1,165 | 4 | 3.4 | 3 | 2.6 |
| 2016 | 17 | 1,988 | 14 | 7.0 | 2 | 1.0 |
| 2017 | 23 | 2,074 | 11 | 5.3 | 8 | 3.9 |
| 2018 | 27 | 1,552 | 15 | 9.7 | 10 | 6.4 |
| 2019 | 30 | 2,206 | 16 | 7.3 | 10 | 4.5 |

straight line repair. The Fisher repair was introduced in 2017. Between 2007 and 2013 the Von Langenbeck palatoplasty technique was the most favoured method of palatal repair. This was subsequently taken over by the Bardach's two-flap palatoplasty technique. Intravelar veloplasty and Furlow's palatoplasty techniques were introduced in the year 2014, Fig 4.

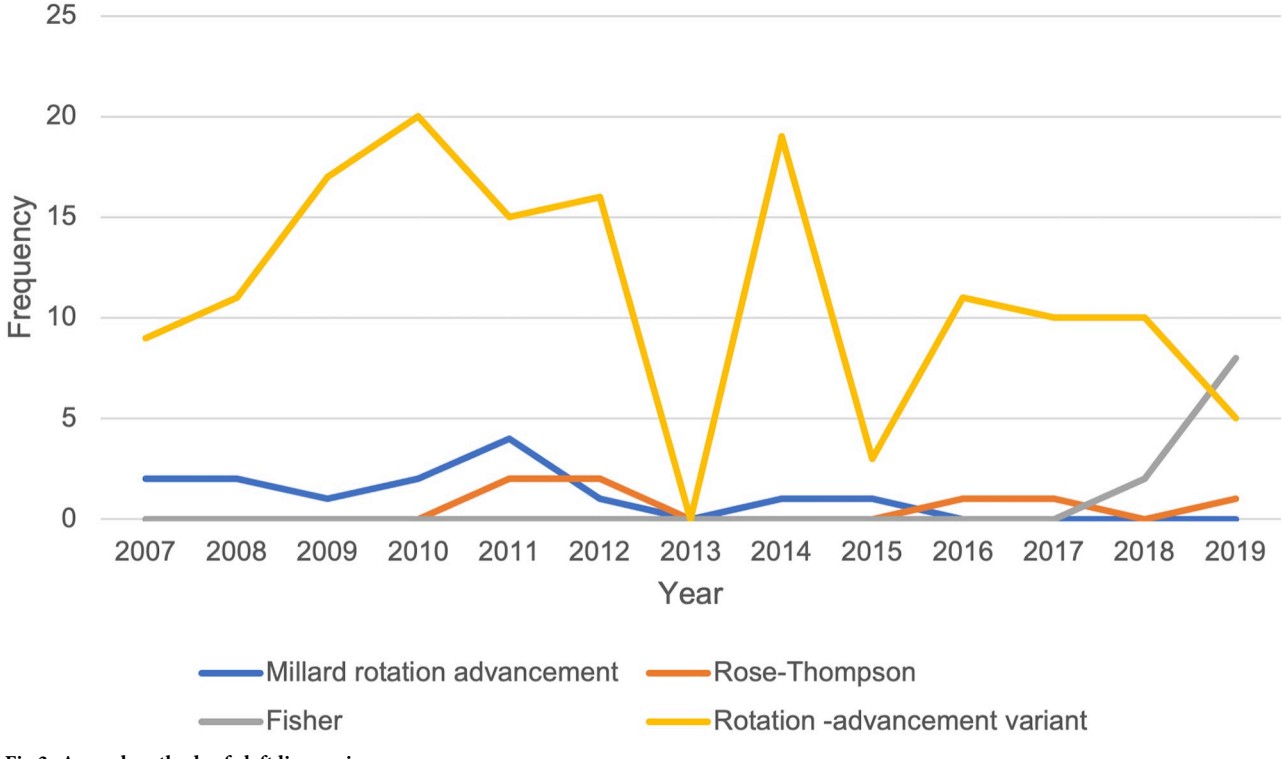

**Fig 3. Annual methods of cleft lip repair.**

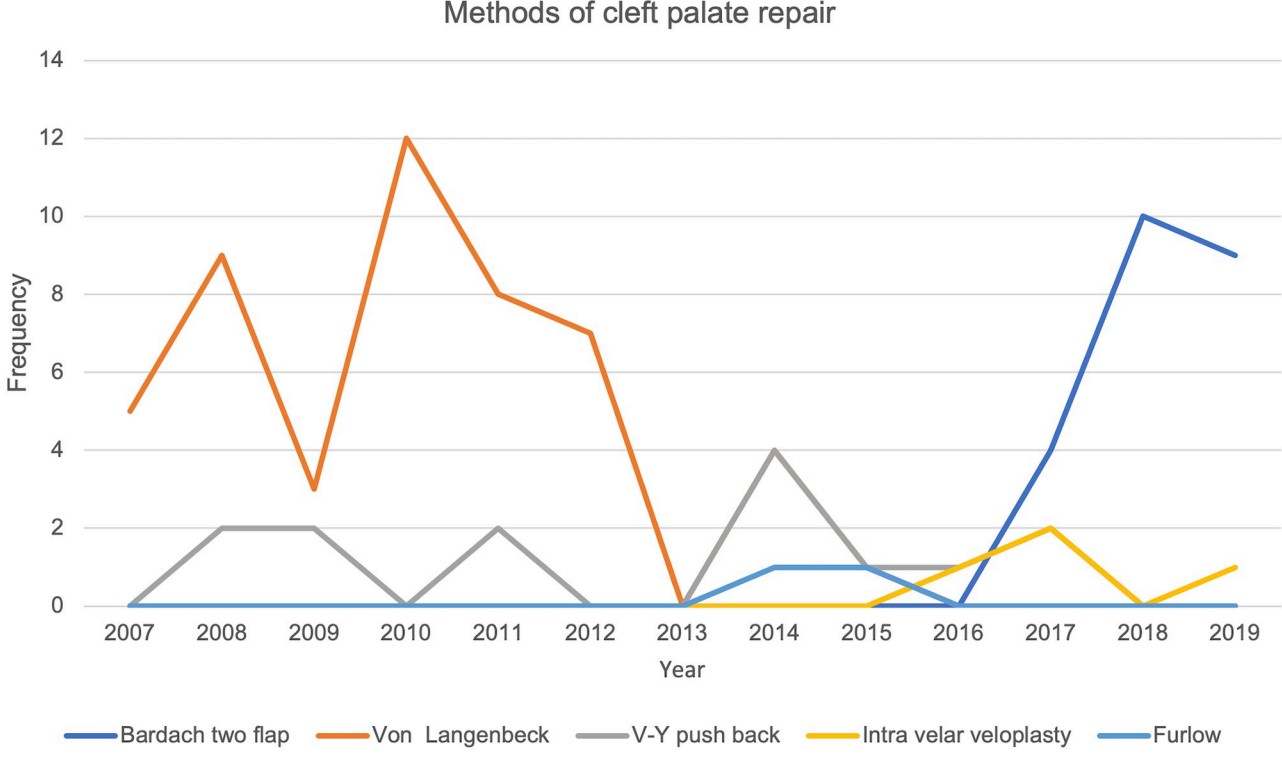

**Fig 4. Annual methods of palatal repair.**

**Association of demographic and clinical parameters with age at primary surgery.**
Nearly all the primary surgeries that were done in children less than 6 months of age were for cleft lip repair as compared with cleft palate repair. Most of the surgeries for cleft palate repair were done in children greater than 18months of age (n = 63, 67%). Based on the severity of the cleft anomaly, children with less severe deformity of the unilateral cleft lip (incomplete cleft) (n = 23,56.1%) had their surgeries later (>18months) than children with the more severe anomaly (complete cleft lip and palate) (n = 3,7.3%), p = 0.001. Similarly, children with unilateral defects had later surgeries compared with those with bilateral defects (4 (97.6%) Vs 1 (2,4%)), p = 0.002. More children with incomplete palatal defects (n = 43, 68%) had later surgeries (>18 months) than children with complete defects (n = 20, 31.8%), p = 0.05. There was no association between gender and age at primary cleft repair. The association between age at primary surgery and the demographic and clinical parameters are shown in Table 3.

## Kaplan Meir plot of the time from birth to primary surgery

The total time of follow-up was 14,770 child-month and the rate of primary cleft surgery was 18 per thousand child-months after birth. Furthermore, the median time from birth to primary surgery was 10months while 25% of the babies had surgery after 48 months. (Fig 5A, Table 4).

There was no statistically significant difference in the survival pattern of time to surgery by gender (logrank P-value = 0.90). Fig 5B, Table 4.

However, the rate of primary surgery was least among babies with cleft lip and palate (14.4 per thousand child-months) followed by the rate of primary surgery among babies with cleft

**Table 3. Association of demographic and clinical parameters with age at primary surgery.**

| Characteristics | Age at Surgery (months) | | | | | P-value |
|---|---|---|---|---|---|---|
| | < 6 | 6–11 | 12–17 | ≥18 | Total | |
| | N (%) | N (%) | N (%) | N (%) | N (%) | |
| Weight (Median, IQR), kg | 5.4 (5.0–6.0) | 7 (5.6–8.4) | 8.2 (6.5–9.0) | 23 (13–48) | 8.0(5.5–16) | 0.0001$ |
| **Gender** | | | | | | |
| Male | 65(55.6) | 25(58.1) | 14(58.3) | 69(53.1) | 173 (55.1) | 0.922£ |
| Female | 52(44.4) | 18 (41.9) | 10(41.7) | 61(46.9) | 141 (44.9) | |
| **Type of repair** | | | | | | |
| Primary | 116 (99.2 | 41 (95.4) | 23 (95.8) | 110 (84.6) | 290 (92.4) | <0.001Fi |
| Secondary | 1 (0.9 | 2 (4.7) | 1 (4.2) | 20 (15.4) | 24 (7.6) | |
| **Type of primary repair** | | | | | | |
| Cleft lip | 111 (60.3) | 23 (12.5) | 8 (4.4) | 42 (22.8) | 184 (100.0) | <0.001 |
| Cleft palate | 1 (1.1) | 16 (17.0) | 14 (14.9) | 63 (67.1) | 94 (100.0) | |
| **Severity of unilateral Cleft** | | | | | | |
| Incomplete cleft lip | 24 (26.3) | 7 (46.7) | 1 (12.5) | 23 (56.1) | 55 (34.5) | 0.001Fi |
| Complete cleft lip | 34 (37.4) | 3 (20.0) | 5 (62.5) | 15 (36.6) | 57 (36.8) | |
| Complete cleft lip and palate | 33 (36.3) | 5 (33.3) | 2 (25.0) | 3 (7.3) | 43 (27.7) | |
| **Severity of cleft palate** | | | | | | |
| Incomplete cleft palate | 0 (0.0) | 14 (87.5) | 13 (92.9) | 43 (68.3) | 70 (74.5) | 0.05Fi |
| Complete cleft palate | 1 (100.0) | 2 (12.5) | 1 (7.1) | 20 (31.8) | 24 (25.5) | |
| **Laterality of cleft lip** | | | | | | |
| Unilateral | 91 (82.0) | 15 (65.2) | 8 (100.0) | 41 (97.6) | 155 (84.2) | 0.002Fi |
| Bilateral | 20 (18.0) | 8 (34.8) | 0 (0.0) | 1 (2.4) | 29 (15.8) | |

$: Kruskal Wallis;

£ Pearson's Ch-square;

FiFisher's exact

lip (20.2 per thousand child-months) and babies with bilateral cleft lip had the highest rate of primary surgery (52.4 per thousand child-months). Furthermore, the median time to surgery was 5 months for both unilateral and bilateral cleft lip surgeries while the median time from birth to primary surgery among babies with both cleft lip and palate was 15 months. Fig 5C.

## Prevalence and predictors of late surgery

Based on the definition of late primary surgery (12 months and beyond for lip repair and 18 months for palate repair) Of the 278 babies that had primary repair, the prevalence of late primary repair was 40.7% (95% CI: 35.0%–46.6%), n = 113/278. From Table 5, there was a statistically significant relationship between the type of cleft and hazard of delay in primary repair. Thus, there was a 22 -fold hazard of late primary cleft repair among babies with unilateral cleft lip as compared to the hazard of late repair among babies with bilateral cleft lip (Adj HR: 22.4, 955 CI: 2.59–193.70, P-value = 0.005). Likewise, there was a 23 -fold hazard of late primary cleft repair among babies with cleft lip and palate as compared to the hazard of late repair among babies with bilateral cleft lip (Adj HR: 23.4, 95%CI: 2.69–202.5, P-value = 0.004).

## Discussion

We set out to determine the surgical trends and pattern of cleft surgeries at UCH, Ibadan from 2007–2019 and evaluated the predictors of late primary cleft surgeries. The period of study

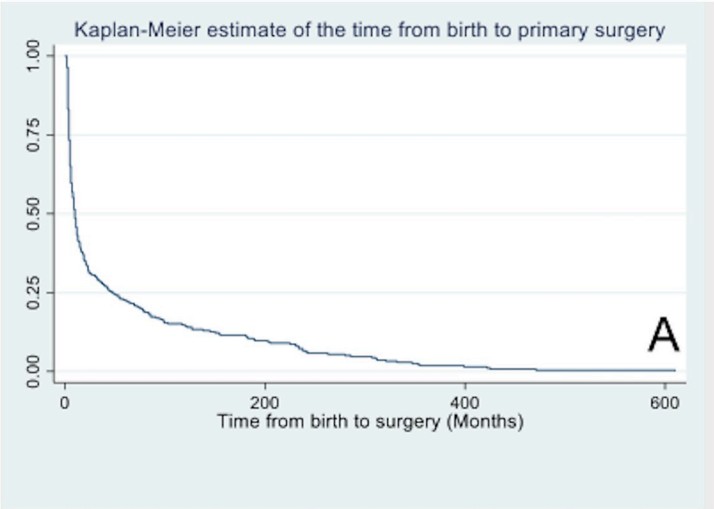

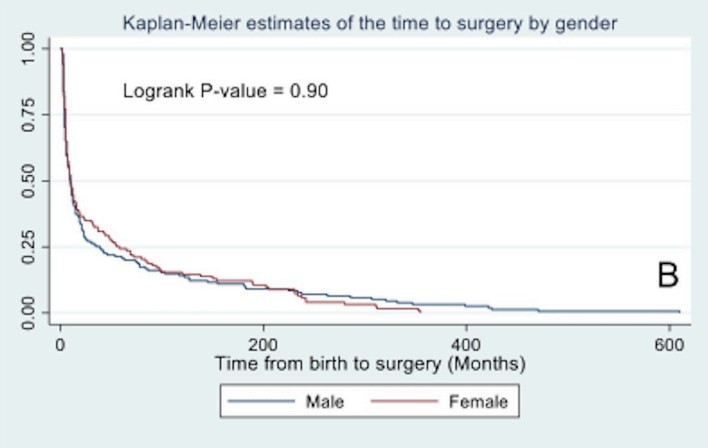

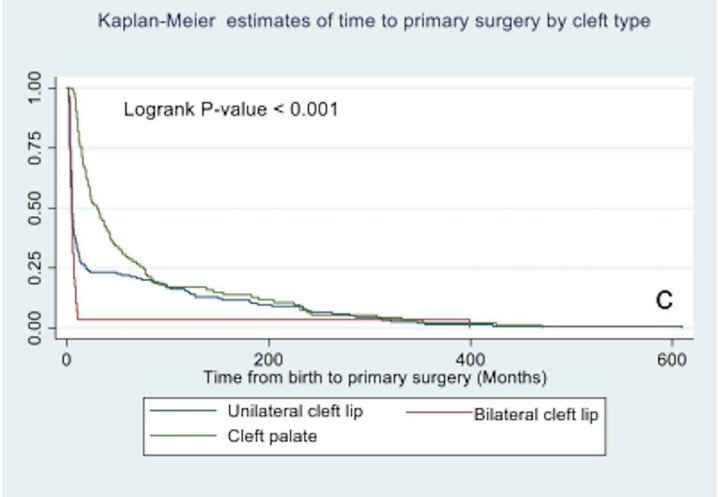

**Fig 5.** A. Kaplan Meir plot of the time to cleft surgeries. B. Kaplan Meir plot of the time to cleft surgeries stratified by gender. C. Kaplan Meir plot of the time to cleft surgeries stratified by type of cleft defect.

**Table 4. Association between survival time and gender and type of primary surgery.**

| Characteristics | Follow-up time (months) | Rate of surgery (per thousand child-month) | Number of babies | Survival time (months) | | | P-value[£] |
|---|---|---|---|---|---|---|---|
| | | | | 25% | 50% | 75% | |
| **Overall** | 14,770 | 18.8 | 278 | 4 | 10 | 48 | |
| **Gender** | | | | | | | |
| Male | 8,339 | 18.6 | 155 | 4 | 10 | 39 | 0.90 |
| Female | 6,431 | 19.1 | 123 | 5 | 10 | 58 | |
| **Weight (Median, IQR), kg** | | | | | | | |
| < 5 | 235 | 144.7 | 34 | 3 | 4 | 10 | |
| ≥5 | 14,535 | 16.8 | 244 | 5 | 11 | 63 | |
| **Type of cleft** | | | | | | | |
| Unilateral Cleft lip | 7,691 | 20.2 | 155 | 3 | 5 | 19 | < 0.001 |
| Bilateral cleft lip | 553 | 52.4 | 29 | 4 | 5 | 7 | |
| Cleft lip and palate | 6,526 | 14.4 | 94 | 15 | 30 | 76 | |

[£:] Logrank test

covered the commencement of the partnership between smile train international and UCH in offering free cleft care to patients. We found a steep rise in the number of surgeries performed from the onset of the partnership attaining a peak about four years afterwards. Nineteen surgeries were performed in 2007 and it increased to a peak of 39 surgeries in 2010. However, there was fluctuation in the annual number of surgeries, but the peak of 2010 was not attained through the subsequent years. This was also mirrored by the incidence of cleft surgeries per 1000 live births. Primary cleft lip repairs were the predominant surgeries performed. Modifications of the Millard rotation advancement flap was the most favoured method of lip repair through the years. For palatoplasty, the Von Langenbeck method was favoured in the earlier 7 years but discontinued and replaced with the Bardach's two-flap palatoplasty in the last 6 years of the study. The less severe anomalies such as incomplete cleft lip, incomplete cleft palate or unilateral defects were significantly associated with late primary repairs. About four out of ten babies in our cohort had late surgeries.

The gender distribution in this study is similar to other studies that have reported an overall male preponderance with female preponderance in Isolated cleft palate [13, 20, 21]. A study

**Table 5. Cox proportional regression of the hazard of late primary repair among babies with cleft lip and palate.**

| Variable | Univariable | | P-value | Multivariable | | P-value |
|---|---|---|---|---|---|---|
| | Hazard ratio | 95% Confidence interval | | Hazard ratio | 95% Confidence interval | |
| **Type of cleft** | | | | | | |
| Bilateral cleft lip | 1.00 | Reference | Reference | 1.00 | Reference | Reference |
| Unilateral Cleft lip | 2.83 | 0.39–20.81 | 0.31 | 22.4 | 2.59–193.70 | 0.005 |
| Cleft lip and palate | 4.04 | 0.55–29.54 | 0.17 | 23.35 | 2.69–202.54 | 0.004 |
| **Gender** | | | | | | |
| Male | 1.00 | Reference | Reference | 0.95 | 0.63–1.42 | 0.805 |
| Female | 1.08 | 0.74–1.59 | 0.42 | | | |
| **Weight (kg)** | 0.86 | 0.84–0.89 | < 0.001 | 0.85 | 0.82–0.88 | < 0.001 |
| < 5 | 1.00 | Reference | Reference | | | |
| ≥5 | 0.10 | 0.04–0.28 | < 0.001 | | | |

from China however observed more males in patients with isolated cleft palate, which they adduced to likely, cultural female discrimination in their setting [14]. Laterality of the anomalies in our study shows a predominance of left sided clefts as reported in other studies [20, 21].

The steep rise in the number of surgeries performed from the onset of the partnership could be because of the increasing awareness of free surgeries, possibly reflecting a mop-up of backlogs of unrepaired clefts. This annual low volume of cleft surgeries has earlier been reported by Olasoji et al. They identified low volume operators in Nigeria that performed not more than ten cleft surgeries in a year [22]. We found that the trend in secondary cleft surgeries were fairly stable although the prevalence was low. This trend is secondary surgeries is contrary to the observation by Purnell et al. they found that the longer (above 6 years) the partnership of hospitals with Smile Train, the higher were the number of secondary surgeries [23].

Primary cleft lip surgeries were the predominant cleft surgeries carried out at our hospital similar to reports from other studies in Nigeria and other low- and middle-income countries [8, 24, 25]. Of these, surgeries on patients with CL were more than those on patients with CLP. However other reviews have reported a preponderance of surgeries on patients with CLP [13, 21]. Primary palatoplasties formed a third of the total cleft surgeries in our series. This was quite similar to the series reported by Olutayo et al from a similar tertiary health facility in southwest Nigeria. We found that few proportion of cleft repairs among our cohort were secondary surgeries. In contrast, Purnell et al [23] reported that secondary surgeries increased with increased length of partnership between partner hospitals and the Smile Train in developing countries. Although, Oginni et al. [26] noted that there was under-reporting of other types of cleft care such as revision surgeries and alveolar bone grafts which are secondary surgeries, the data that was utilised for this present study was comprehensive and captured all the various types of cleft surgeries that were performed during the study period.

Considering that almost ninety percent of our surgeries were primary surgeries the mean age of five years at primary surgeries of this cohort of cleft patients is high. A study from Northern Nigeria reported a mean age of 12.4 yeas in their cohort of 149 patients with cleft anomalies [27]. A Ugandan study [28] reported a mean age of three years while the study by Conway et al. that looked at cleft surgeries across Africa reported a mean age of nine years [13]. A more recent study from southwest Nigeria reported a mean age of 2.5 years at primary lip repair [25]. From China an average age of 1.8 years for lip repair and 5.9 years for palatal repair was seen in the study by Kling et al [14]. These contrast with studies from developed countries where timely repair is the norm [2, 12]. Reasons for late repair of clefts in developing countries and underserved areas of high-income countries are reportedly multifaceted and include low socioeconomic status, lack of access to care, ignorance, and lower educational status of parents of children with clefts [12, 29].

The Millard rotation advancement flap repair was the most employed method of repair for the unilateral cleft lip. This finding was like other studies in our sub region and Africa. [22, 25, 30, 31]. Patel and Patel however reported that the aesthetic outcomes after the Fisher method of cleft lip repair were maintained irrespective of the cleft severity as compared with the Millard rotation advancement method. The Fisher method was introduced in the last two years of this study [32]. Superior results of the Fisher repair were also reported by ElMaghraby et al [33]. Although the Von Langenbeck palatoplasty was the method more commonly employed for closure of cleft palate, the trend showed that this method had been completely abandoned in the last six years of the study period and replaced with the two flap palatoplasty technique by Bardach. This may probably reflect a change in surgeon dynamics and knowledge of palatoplasty techniques as well as the advantages the Bardachs's palatoplasty has over Von Langenbeck palatoplasty. In a systematic review and meta-analysis by Stein et al they found less

fistulae associated with the Furlow repair when compared with the Von Langenbeck andVeau/Wardil/Kilner palatoplasty techniques. Additionally. Velopharyngeal insufficiency was less in Bardach's palatoplasty when compared with Von Langenbeck palatoplasty [34]. Von Langenbeck palatoplsty was however still more commonly used in other similar institutions in Nigeria and amongst cleft surgeons [15, 25, 30]. The modified Tennison-Randall technique for unilateral cleft lip repair, Bardach's and Von Langenbeck's methods were favoured in an institution in Italy [35]. The rotation- advancement technique, pushback and double opposing z platies were commoner in Korea [36]. In the United States of America (USA), the Fisher method for unilateral cleft lip repair is gaining popularity over the rotation-advancement techniques. Two-flap palatoplasty with intravelar veloplasty is reported to produce superior speech outcomes [37]. This combination is favoured for palatal repair in the USA [38, 39]. The reasons why surgeons in Nigeria still stick to the older traditional methods of cleft lip and palate repair needs further exploration.

## Prevalence and hazards of late repairs

Cleft surgery of any sort should be individualized based on the objectives of the surgery and the clinical condition of the patient. Nevertheless, various guidelines have been put forward to optimize safety, ensure adequacy of repair, and improve function while limiting morbidity to the patient with the cleft deformity. Therefore, surgical procedures should be well planned to reduce exposure to anaesthesia from multiple surgeries [3]. One of the common guidelines for the timing of surgical cleft lip repair is the rule of ten [4]. Although its validity is being put to question, it is still widely used [8–10]. This rule is also employed in our institution. The ACPA introduced parameters to evaluate and treat patients with cleft lip and palate [40]. This earlier edition of ACPA guidelines recommended that lip repair be completed by six months and palatal repair by 18 months while the latest revision provides that initiation of lip repair be commenced before 12 months and palatal repair completed by 18 months [3]. They further advised that earlier surgeries are desired if the conditions are right. Early repairs are aimed at improving appearance, speech, hearing, psychosocial development and avoiding impediments to social integration. Based on the ACPA guidelines, Cassell et al. from a resourced nation, reported nearly 90% of the patients in their study had had their surgeries by the age of six months [12]. In this study we found that the prevalence of late primary cleft lip repair was about a third of the patients having primary cleft lip surgery while the prevalence of late palatal repair was more than two thirds of those who received primary palatoplasty. Therefore approximately 60% of those who had cleft lip repair and less than 30% of those who had cleft palate repair met the guidelines as recommended by the ACPA. Despite the availability of free cleft surgeries provided by Smile Train, late repairs are prevalent in our setting [15, 25]. Interestingly we found that the nature of the deformity was associated with the prevalence of late primary cleft repair. Unilateral cleft lip defects had a higher prevalence of late primary repair than bilateral defects while incomplete clefts of the palate had a higher prevalence of late primary repair than complete defects. Cassell et al. found no association between the type of cleft and timeliness of the repair in their study from North Carolina [12]. We could not find studies that have investigated this relationship in our setting. The lower prevalence of late primary repair in bilateral defects may reflect the perceived severity of the defects by the parents and the need to seek care to avoid stigmatization. Incomplete defects of the palate are readily recognized, and patients may not present until the speech deficit associated with the deformity becomes apparent. Cassel et al. identified predictors of timely repair to be receipt of maternal care services, access to care and race. Blacks were more likely to have late repairs for orofacial clefts [12]. We also found that amongst children who had cleft lip, incomplete defects had a

significantly higher prevalence (44%) of late primary repair when compared with the more severe complete cleft lip and palate (12%). This finding was also mirrored by the palatal defects. Incomplete clefts of the palate had a higher prevalence (68.3%) of late primary repair than complete clefts of the palate (31.8%). Compared with children who had bilateral cleft lip, children with unilateral cleft lip had a significantly increased risk of late primary repair.

## Limitations of the study

The data base used for this study was limited in variables such as education of the parent s which may also have affected the late primary repair. The nature of the data base used for this research does not allow for relating the changing trend in the methods of cleft lip and palate repair to outcomes of repair.

## Conclusions and recommendations

The trends in number of cleft surgeries show a low volume in later years. There has ben a change from Von Langenbeck palatoplasty to Bardachs two-flap palatoplasty. Intra-velar velo-plasty and Fisher's method of lip repair were introduced in later years. There was a higher risk of late primary repair in children with unilateral cleft lip. There is a need to investigate the reason for the changing trends in techniques of cleft surgeries and determine the impact of this on the functional outcomes of the repairs. More attention would need to be paid to children with less severe cleft deformities to ensure they have early repairs. Increased awareness on identification of incomplete clefts of the palate may increase their recognition and early presentation.

## Acknowledgments

We acknowledge the National Surgical, Obstetric, Anaesthesia and Nursing Plan (NSOANP)/ Smile Train partnership for fostering the collaboration for this work.

## Author Contributions

**Conceptualization:** Afieharo Igbibia Michael, Gbenga Olorunfemi.

**Data curation:** Afieharo Igbibia Michael, Gbenga Olorunfemi, Adeola Olusanya.

**Formal analysis:** Afieharo Igbibia Michael, Gbenga Olorunfemi.

**Investigation:** Afieharo Igbibia Michael.

**Methodology:** Afieharo Igbibia Michael, Gbenga Olorunfemi.

**Project administration:** Afieharo Igbibia Michael, Adeola Olusanya, Odunayo Oluwatosin.

**Software:** Gbenga Olorunfemi.

**Supervision:** Afieharo Igbibia Michael, Odunayo Oluwatosin.

**Validation:** Afieharo Igbibia Michael, Gbenga Olorunfemi, Adeola Olusanya, Odunayo Oluwatosin.

**Visualization:** Afieharo Igbibia Michael.

**Writing – original draft:** Afieharo Igbibia Michael, Gbenga Olorunfemi.

**Writing – review & editing:** Afieharo Igbibia Michael, Gbenga Olorunfemi, Adeola Olusanya, Odunayo Oluwatosin.

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
