## [Decision Letter · Decision Letter 0]

20 Sep 2022

PONE-D-22-24340Trends of cleft surgeries and predictors of late primary surgery among children with cleft lip and palate at the University College Hospital, Nigeria: a retrospective cohort studyPLOS ONE

Dear Dr. Michael,

Thank you for submitting your manuscript to PLOS ONE. After careful consideration, we feel that it has merit but does not fully meet PLOS ONE’s publication criteria as it currently stands. Therefore, we invite you to submit a revised version of the manuscript that addresses the points raised during the review process.

We look forward to receiving your revised manuscript.

Kind regards,

Johari Yap Abdullah, B.S. & I.T, GradDip ICT, M.Sc, Ph.D.

Academic Editor

PLOS ONE

Journal Requirements:

3. Please amend your current ethics statement to address the following concerns:

a) Did participants provide their written or verbal informed consent to participate in this study?

5. Please upload a new copy of Figures 2,3,4,5A,5B and 5C as the detail is not clear. Please follow the link for more information: https://blogs.plos.org/plos/2019/06/looking-good-tips-for-creating-your-plos-figures-graphics/" https://blogs.plos.org/plos/2019/06/looking-good-tips-for-creating-your-plos-figures-graphics/

Reviewers' comments:

Reviewer's Responses to Questions

**Comments to the Author**

1. Is the manuscript technically sound, and do the data support the conclusions?

Reviewer #1: Partly

Reviewer #2: Yes

2. Has the statistical analysis been performed appropriately and rigorously? 

Reviewer #1: Yes

Reviewer #2: Yes

3. Have the authors made all data underlying the findings in their manuscript fully available?

Reviewer #1: Yes

Reviewer #2: Yes

4. Is the manuscript presented in an intelligible fashion and written in standard English?

Reviewer #1: No

Reviewer #2: No

5. Review Comments to the Author

Reviewer #1: Dear Authors

1.Tertiary Hospital or Hospital University usually have their own " cleft team" which comprises of surgeons, orthodontist, geneticist, audiologist, dentist and others. Is there any cleft team was formed at your hospital? as this will offers a comprehensive treatments and follow up to the patients.

The hazard of delay in primary repair of cleft are not mentioned in the manuscript although statistical analysis of relationship between the types of cleft and hazard of delay in primary repair was mentioned. What are the hazard involved?

This study is a retrospective study, there were a significant data from follow up sessions that can support the changing trends of surgical procedures, with the supported data, authors can conclude the reasons of the changes of surgical trends whether it is based on surgical defects or observable complications and advantages of that particular surgical techniques.

"The nature of the data base used for this research

does not allow for relating the changing trend in the methods of cleft lip and palate repair to

outcomes of repair." Even though authors clearly state the limitation, to absolutely determined the factors that change the surgical approaches, the data related to preoperative assessment, postoperative follow up to compare which surgical approaches causes less complications must be included if the authors really wanted to determine the changing trends of cleft surgical procedures.

For the patients with cleft lip and palate treated at your center, any pre-surgical orthodontic treatment such as Nasoalveolar Molding( NAM) appliances were offered to patients as this will determine surgical approaches that will be provided by the surgeons.

The second and third objectives of this study were well elaborated by the authors.

2. The statistical anaylsis for this study has been performed appropriately.

3. All data underlying the findings are fully available in the manuscript.

4. There are a lot of spelling errors particularly the surgical procedures.

Abstract

Subheading

Result

a) Bardachs two flap palatoplasty has replaced Von Lagenbeck palatoplasty as the

commonest method of palatal repair- Von Langenbeck

Conclusion

a)There has been a change from Von Lagenbeck palatoplasty to Bardachs two-flap palatoplasty- Von Langenbeck

Introduction

Subheading

Trends in methods of cleft lip and palate repair

Spelling error of Von Langenbeck

the "Von Lagenbeck" procedure should be corrected for each of the mentioned sentences to " Von Langenbeck"

Subheading Discussion

Spelling error

"Although the Von lagenbecks palatoplasty was method more commonly employed

for closure of cleft palate the trend showed that tis method had been completely abandoned in the

last six years of the study period and replaced with the two flap palatoplasty technique by

Bardach. This may probably reflect a change in surgeon dynamics and knowledge oge of

palatoplasty techniques as well as the advantages the Bardachs’s palatoplasty has over Von

Lagenbacks palatoplasty"

Surgical technique and tis should be change to this.

Knowlegde oge to knowledge of.

please consider to proof read your manuscript before the submission.

Reviewer #2: The manuscript is well written, need a minor revision.

1. Surgical procedures should be well planned to reduce exposure to anaesthesia from multiple surgeries (3) (One reference is not sufficient to support this statement. Add few more references. The different surgical techniques must also be included)

2. The procedure of obtaining informed consent is not mentioned. Kindly elaborate the informed consent procedure (Study design and setting)

3. Only primary surgeries of typical clefts were utilized for bivariate and multivariable a analysis to determine the primary outcomes, as such atypical clefts and secondary surgeries were not analyzed further (Rewrite the sentences)

4. Further explanation is needed in terms of advancement in latest trends of cleft surgeries. For example, Bardach's technique and other techniques must need a detailed explanation. (Add a new paragraph before Prevalence and hazards of late repairs section.

5. Reference format is not proper. Please refer to ICMJE format for standard journal article

6. The figures must be uploaded in 300 dpi format. (Refer to author guidelines for figures)

6. PLOS authors have the option to publish the peer review history of their article (what does this mean?). If published, this will include your full peer review and any attached files.

Reviewer #1: No

Reviewer #2: **Yes: **Mohamed Zahoor Ul Huqh

---

## [Author Response · Author response to Decision Letter 0]

14 Nov 2022

Academic editor-

Please ensure that your manuscript meets PLOS ONE's style requirements, including those for file naming 

Response- The manuscript has been formatted to meet PLOS ONE’s style requirements 

 You indicated that you had ethical approval for your study. In your Methods section, please ensure you have also stated whether you obtained consent from parents or guardians of the minors included in the study or whether the research ethics committee or IRB specifically waived the need for their consent 

Response- to the retrospective nature of the study the ethics committee waived the need for consent. Methods. Study design and setting. Page 7, lines 154,155 159-160

Please amend your current ethics statement to address the following concerns:

a) Did participants provide their written or verbal informed consent to participate in this study?

 Response- A written informed consent was obtained from all adult patients and parents of children for surgery, entry of their data into the database and utilization for research. Further need for consent for this study was waived by the ethics committee due to the retrospective nature of the study. Methods. Study design and setting. Page 7, lines 154,155 159-160

In your Data Availability statement, you have not specified where the minimal data set underlying the results described in your manuscript can be found. PLOS defines a study's minimal data set as the underlying data used to reach the conclusions drawn in the manuscript and any additional data required to replicate the reported study findings in their entirety. All PLOS journals require that the minimal data set be made fully available. 

Response-The database that utilized was the database of Smile Train with some restrictions to data availability. Data will be made available on request to the data manager at stxadmin@smiletrain.org

 Please upload a new copy of Figures 2,3,4,5A,5B and 5C as the detail is not clear. 

Response- Clearer figures 2,3,4,5A,5B and 5C have been uploaded Figures

Please review your reference list to ensure that it is complete and correct

.

Response-All the references have been reviewed to comply with PLOS ONE format References 

Is there any cleft team was formed at your hospital? as this will offers a comprehensive treatment and follow up to the patients.

 Response-Our hospital does not have the full complement of all expected members of a cleft team. The cleft team comprises of cleft surgeons, clinical and public health nurses and a nutritionist. 

The hazard of delay in primary repair of cleft are not mentioned in the manuscript although statistical analysis of relationship between the types of cleft and hazard of delay in primary repair was mentioned. What are the hazard involved?

Response- We obtained hazard ratio from the Cox proportional hazard regression modelling, and this was reported in Table 5. The hazard referred to is the hazard ratio 

Even though authors clearly state the limitation, to absolutely determined the factors that change the surgical approaches, the data related to preoperative assessment, postoperative follow up to compare which surgical approaches causes less complications must be included if the authors really wanted to determine the changing trends of cleft surgical procedures.

Response-We thank the reviewer for this query which is a limitation of the study as stated in the manuscript. The nature of the database used for the study does not allow for this additional information. This study nevertheless has encouraged us to put in a more structured follow up mechanism for a prospective study. 

For the patients with cleft lip and palate treated at your center, any pre-surgical orthodontic treatment such as Nasoalveolar Molding( NAM) appliances were offered to patients as this will determine surgical approaches that will be provided by the surgeons.

Response- None of the patients had Nasoalveolar moulding. There was no orthodontist on the team within the period of the study 

Spelling errors

Von Lagenbeck 

Response-This has been changed to Von Langenbeck Abstract. Subheading, Result. Page 3, line 67

Von Lagenbeck 

Response-This has been changed to Von Langenbeck Abstract-Conclusion. Page 4, line 79

Von Lagenbeck 

Response-This has been changed to Von Langenbeck Results. Subheading, Trends in methods of cleft lip and palate repair. Page 14, lines 248-249

Although the Von lagenbecks palatoplasty was method more commonly employed

for closure of cleft palate the trend showed that tis method had been completely abandoned in the

last six years of the study period and replaced with the two flap palatoplasty technique by

Bardach. This may probably reflect a change in surgeon dynamics and knowledge oge of

palatoplasty techniques as well as the advantages the Bardachs’s palatoplasty has over Von

Lagenbacks palatoplasty" 

Response-Von Lagenbeck has been changed to Von Langenbeck, tis has been changed to this. Knowlegde oge has been changed to knowledge of. Discussion. Page 23, line 340. Page 25, line 411. Page 25, line 413

Surgical procedures should be well planned to reduce exposure to anaesthesia from multiple surgeries (3) (One reference is not sufficient to support this statement. Add few more references. The different surgical techniques must also be included) 

Response-More references have been added and the different surgical techniques included. Introduction -Page 5, lines 91-95

The procedure of obtaining informed consent is not mentioned. Kindly elaborate the informed consent procedure (Study design and setting)

Response-This had been added to the methods section Methods-Page 7, lines 154,155 159-160

Only primary surgeries of typical clefts were utilized for bivariate and multivariable a analysis to determine the primary outcomes, as such atypical clefts and secondary surgeries were not analyzed further.

Response- This sentence has been rewritten as advised Methods-Page 8, 181-183

Further explanation is needed in terms of advancement in latest trends of cleft surgeries. For example, Bardach's technique and other techniques must need a detailed explanation. (Add a new paragraph before Prevalence and hazards of late repairs section.

Response- Additional detail on the surgeries have been added as rightfully advised. These were however embedded into the discussion rather than a separate paragraph in order to maintain the flow of thought/discussion. Discussion Page 25, lines 406-409. Pages 25-26, lines 414-475

Reference format is not proper. Please refer to ICMJE format for standard journal article. 

Response- All the references have been revised to comply with Plos One and ICMJE format

The figures must be uploaded in 300 dpi format. (Refer to author guidelines for figures)

Response-The figures have been saved and uploaded in 300dpi format Figures

---

## [Editor Report · Decision Letter 1]

28 Nov 2022

Trends of cleft surgeries and predictors of late primary surgery among children with cleft lip and palate at the University College Hospital, Nigeria: a retrospective cohort study

PONE-D-22-24340R1

Dear Dr. Michael,

We’re pleased to inform you that your manuscript has been judged scientifically suitable for publication and will be formally accepted for publication once it meets all outstanding technical requirements.

Kind regards,

Johari Yap Abdullah, B.S. & I.T, GradDip ICT, M.Sc, Ph.D.

Academic Editor

PLOS ONE
---

## [Editor Report · Acceptance letter]

6 Dec 2022

PONE-D-22-24340R1 

Trends of cleft surgeries and predictors of late primary surgery among children with cleft lip and palate at the University College Hospital, Nigeria: a retrospective cohort study 

Dear Dr. Michael:

I'm pleased to inform you that your manuscript has been deemed suitable for publication in PLOS ONE. Congratulations! Your manuscript is now with our production department. 

Kind regards, 

on behalf of

Dr. Johari Yap Abdullah 

Academic Editor

PLOS ONE